# Effects of non-aversive versus tail-lift handling on breeding productivity in a C57BL/6J mouse colony

**Margaret A. Hull**[1]*, **Penny S. Reynolds**[2], **Elizabeth A. Nunamaker**[1¤]

1 Animal Care Services, University of Florida, Gainesville, Florida, United States of America, 2 Department of Anesthesiology; Statistics in Anesthesiology Research (STAR) Core, College of Medicine, University of Florida, Gainesville, Florida, United States of America

¤ Current address: Global Animal Welfare and Training, Charles River Laboratories, Wilmington, Massachusetts, United States of America

* ma.hull@ufl.edu

**Data Availability Statement:** All data files are available from Harvard Dataverse (https://doi.org/10.7910/DVN/JQENJN).

## Abstract

Non-aversive handling is a well-documented refinement measure for improving rodent welfare. Because maternal stress is related to reduced productivity, we hypothesized that welfare benefits associated with non-aversive handling would translate to higher production and fewer litters lost in a laboratory mouse breeding colony. We performed a randomized controlled trial to examine the effects of a standard method of handling (tail-lift with forceps) versus non-aversive handling with transfer tunnels ('tunnel-handled') on breeding performance in 59 C57BL/6J mouse pairs. Intervention assignments could not be concealed from technicians, but were concealed from assessors and data analyst. An operationally significant effect of tunnel-handling (large enough differences to warrant programmatic change) was defined before study initiation as a 5% increase in productivity, or one extra pup over the reproductive lifetime of each pair. Pairs were randomly allocated to handling intervention and cage rack location, and monitored over an entire 6-month breeding cycle. For each group, we measured number of pups born and weaned, and number of entire litters lost prior to weaning. Differences between transfer methods were estimated by two-level hierarchical mixed models adjusted for parental effects and parity. Compared to tail-lift mice, tunnel-handled mice averaged one extra pup per pair born (+1.0; 95% CI 0.9, 1.1; P = 0.41) and weaned (+1.1, 95% CI 0.9, 1.2; P = 0.33). More tunnel-handled pairs successfully weaned all litters produced (13/29 pairs, 45% vs 4/30 pairs, 13%; P = 0.015), averaged fewer litter losses prior to weaning (11/29 pairs [38%] vs 26/30 pairs [87%]; P <0.001), and had a 20% lower risk of recurrent litter loss. The increase in numbers of pups produced and weaned with tunnel handling met threshold requirement for operational significance. These data and projected cost savings persuaded management to incorporate tunnel handling as standard of care across the institution. These data also suggest that overlooked husbandry practices such as cage transfer may be major confounders in studies of mouse models.

**Funding:** The authors received no specific funding for this work.

**Competing interests:** The authors have declared that no competing interests exist.

## Introduction

The standard method of transferring mice between cages is by lifting animals by the tail. In North American animal facilities, mice are picked up by their tail, oftentimes by using padded-tip forceps to grasp the base of the tail [1]. Handling rodents by the tail both induces and increases stress responses, and animals do not habituate to this method of handling [1–5]. In contrast, non-aversive handling methods (using plastic tubes, tunnels, or cupped hands) reduce anxiety-like behaviors [1–4, 6–12], stress biomarker levels [6, 8, 13]), and anhedonia-like states [5, 6, 8, 12]. Stress reduction effects persist even if exposure to non-aversive handling is brief [3, 9] and animals are further restrained by scruffing and/or subjected to painful procedures, such as intraperitoneal injections [3, 4, 10, 14]. The evidence that non-aversive handling is a major welfare refinement has resulted in implementation of non-aversive handling methods as standard husbandry practice in the United Kingdom [4]. However, to date these methods are rarely employed on a programmatic basis in North America.

There is little information on the effect of non-aversive handling on breeding mice. One study showed that there was no difference between non-aversive handling methods on weaning success. However, the study was designed to evaluate effects on husbandry efficiency, and a direct comparison of tail-lift handling with tunnel handling on breeding performance *per se* was not assessed [1]. Because non-aversive handling reduces stress in general, it is reasonable to predict it will also result in positive welfare benefits to breeding mice. In general, stress has been shown to negatively affect breeding productivity indices such as pup survivability, robustness, and litter size [15–18]. Female mice have a greater response to stressors than do males [18]; maternal stress reduces oocyte implantation rates [15, 18, 19] and oocyte development potential [19–21], and is associated with reduced pup weight and pup survivability [18, 22, 23]. Stress during pregnancy in laboratory mice can result in smaller litter sizes, either through reduced implantation rates [24, 25], increased pre-weaning mortality [15, 26], or both. Both acute and chronic stress disrupt maternal behavior in rodents (e.g. more fragmented grooming and nursing, increased latency to initiate nursing, more time spent away from pups, reduced pup retrieval, etc.) [27].

Because maternal stress is related to poorer productivity, we hypothesized that the positive welfare benefits associated with NAH of C57BL/6J mice during routine husbandry would translate to higher production and fewer litters lost prior to weaning, compared to animals handled by tail-lift with forceps. We found that effects of handling method alone resulted in a modest increase in pup production and a substantial reduction in pre-weaning litter mortality. However, these results have implications beyond welfare and economics of breeding colony operations. C57BL/6J is a popular and versatile mouse strain used as a research model for various pathological and physiological conditions, as a genetic background strain, and for construction of targeted transgenic lines. Results from this study also show how an often-overlooked husbandry practice such as cage transfer method can have major effects on the phenotype and behavior of a ubiquitous and otherwise well-documented mouse model.

## Results

Monitoring for the entire 6-month follow-up period was completed for 59 breeding pairs (29 tunnel-handled, 30 tail-lift with forceps) by March 31st, 2021, the designated end of the trial. Summary data for both groups are presented in Table 1. A total of 1950 pups in 286 litters were produced, and 1518 pups were weaned, for an overall weaning success rate of 78%. Tunnel-handled pairs produced approximately two more pups born (Fig 1A) and one more pup weaned (Fig 1B), compared to pairs handled by tail lift-with forceps (Table 1). Number of pups both born and weaned declined with parity, with approximately 24% of all pups in both

**Table 1. Summary statistics for C57BL/6J mouse breeding pair productivity by handling method (non-aversive handling with tunnels vs tail-lift with forceps).**

| | Tunnel-handled | Tail-lift with forceps |
|---|---|---|
| Number of breeding pairs | 29 | 30 |
| Total number of litters produced | 141 | 145 |
| Total number of pups born | 1006 | 944 |
| Total number of pups weaned | 792 | 726 |
| Number of entire litters disappeared | 24 | 34 |
| Number of pups euthanized | 11 | 8 |
| Number of litters with ≥1 pup euthanized | 5 | 5 |
| Dam euthanized for dystocia/found dead | 1 | 2 |
| Number of pairs non-productive for >60 d | 5 | 6 |
| Weaning success per group (%) | 79 | 77 |
| Median total pups born per pair (IQR) | 34 (26, 39) | 32 (23, 36) |
| Median total pups weaned per pair (IQR) | 25 (22, 31) | 24 (18, 28) |
| Median inter-litter interval (IQR, days) | 44 (41, 50) | 43 (40, 50) |

groups produced in first litters (S2.5 Table 1 in S2 File). Inter-litter intervals were 1 day longer for tunnel handled pairs compared to tail-lift with forceps (Table 1). When adjusted for parental effects and parity (S2.5 Table 2 in S2 File), tunnel handled pairs averaged one extra pup per pair born (+1.04, 95% CI: 0.95, 1.14; P = 0.41), and weaned (+1.07, 95% CI 0.94, 1.20; P = 0.33).

Fourteen of 29 tunnel-handled pairs (48%) successfully weaned all litters produced, compared to 4/30 (13%) of tail-lift with forceps handled pairs (|z| = 2.428, P = 0.015) (Fig 2). Fifty-eight litters out of 287 litters produced (~20%) disappeared prior to weaning. Loss of one or more complete litters by a given pair was significantly associated with handling method (likelihood ratio $\chi^2$ = 13.48; P = 0.004). Of these, litter loss resulted from three dams (1 tunnel-handled, 2 tail-lift with forceps) having been found dead or euthanized for dystocia. For the remainder, tunnel-handled pairs lost fewer entire litters (23/140 litters, 17%) compared to tail-lift with forceps pairs (32/143 litters, 23%). The relative risk was 0.73 (95% CI: 0.45, 1.19). There was an approximate 20% reduction in risk of recurrent, or repeated, litter loss with NAH (hazard ratio = 0.8; 95% CI 0.5, 1.4; P = 0.2). Five tunnel-handled pairs and six tail-lift pairs exhibited prolonged non-productivity (no litter produced >60 days past the last litter weaned), and were removed from the study per previously established exclusion criteria.

## Discussion

Previous studies have shown that stress reduction associated with non-aversive handling for routine husbandry has demonstrable and long-lasting effects on behavior, stress biomarker levels, and ease of handling [2–4, 9]. Data from this study suggest that tunnel-handling of gravid and nursing mice may also show small but consistent improvements in productivity during routine colony production and breeding operations. In this study, an operationally-significant change in productivity was defined as an increase of 5% (or one extra pup per pair over the reproductive lifetime), which was determined *a priori* to be sufficient evidence of benefit to justify transition from tail-lift with forceps to non-aversive methods for all mouse facilities at this institution. In comparison to tail-lift with forceps pairs, tunnel-handled pairs met the operationally significant threshold of one extra pup born and weaned per pair. They also showed a 7% increase in the proportion of litters successfully weaned per pair. Tunnel-handled

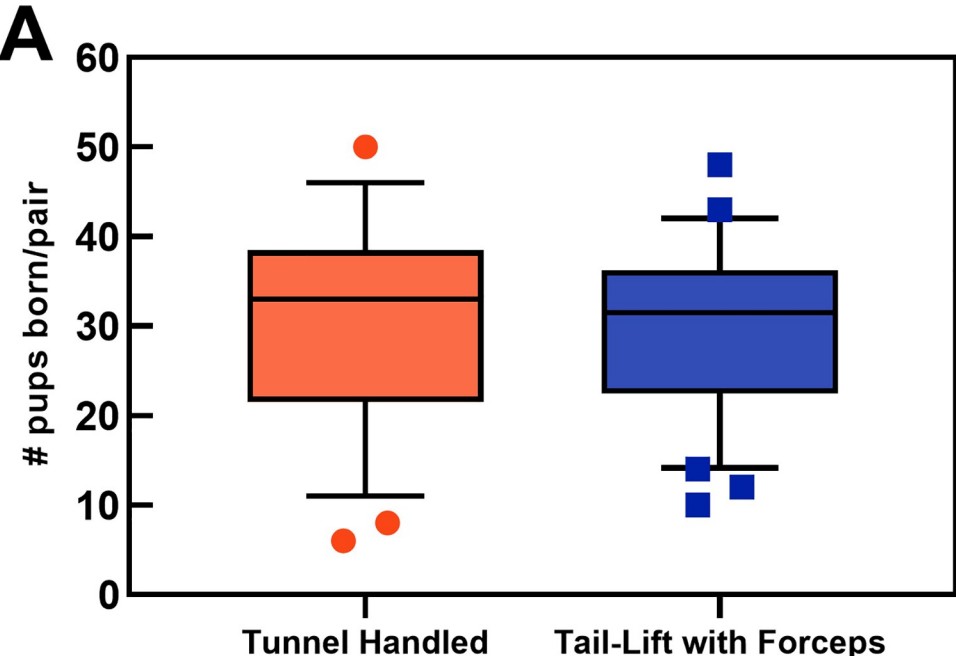

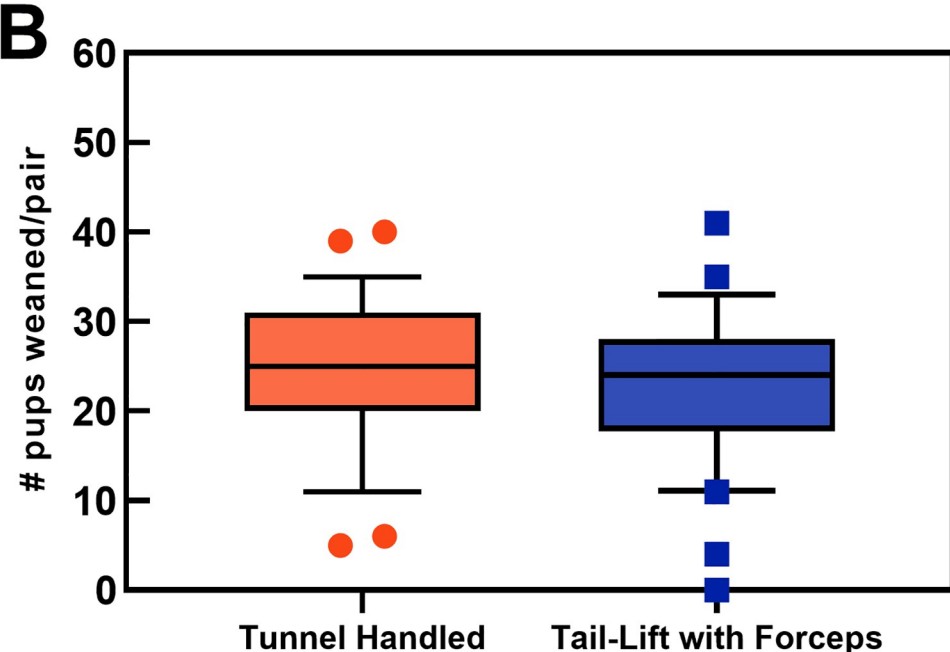

**Fig 1. Distributions of pups born and weaned per pair with tunnel handling (n = 29 pairs) vs tail-lift with forceps (n = 30 pairs).** Frequency distribution and boxplots. (A) Total number of pups born per pair. Median difference (solid black line) was 2 pups more per pair with tunnel handling. (B) Total pups weaned per pair. Median difference (solid black line) was one more pup weaned per pair with tunnel handling.

pairs were less likely to lose all pups in a litter prior to weaning, and had a lower risk of recurrent litter loss.

These data suggest that welfare benefits associated with tunnel-handling primarily accrue through reduction in preventable pup deaths prior to weaning. Losses of complete litters may

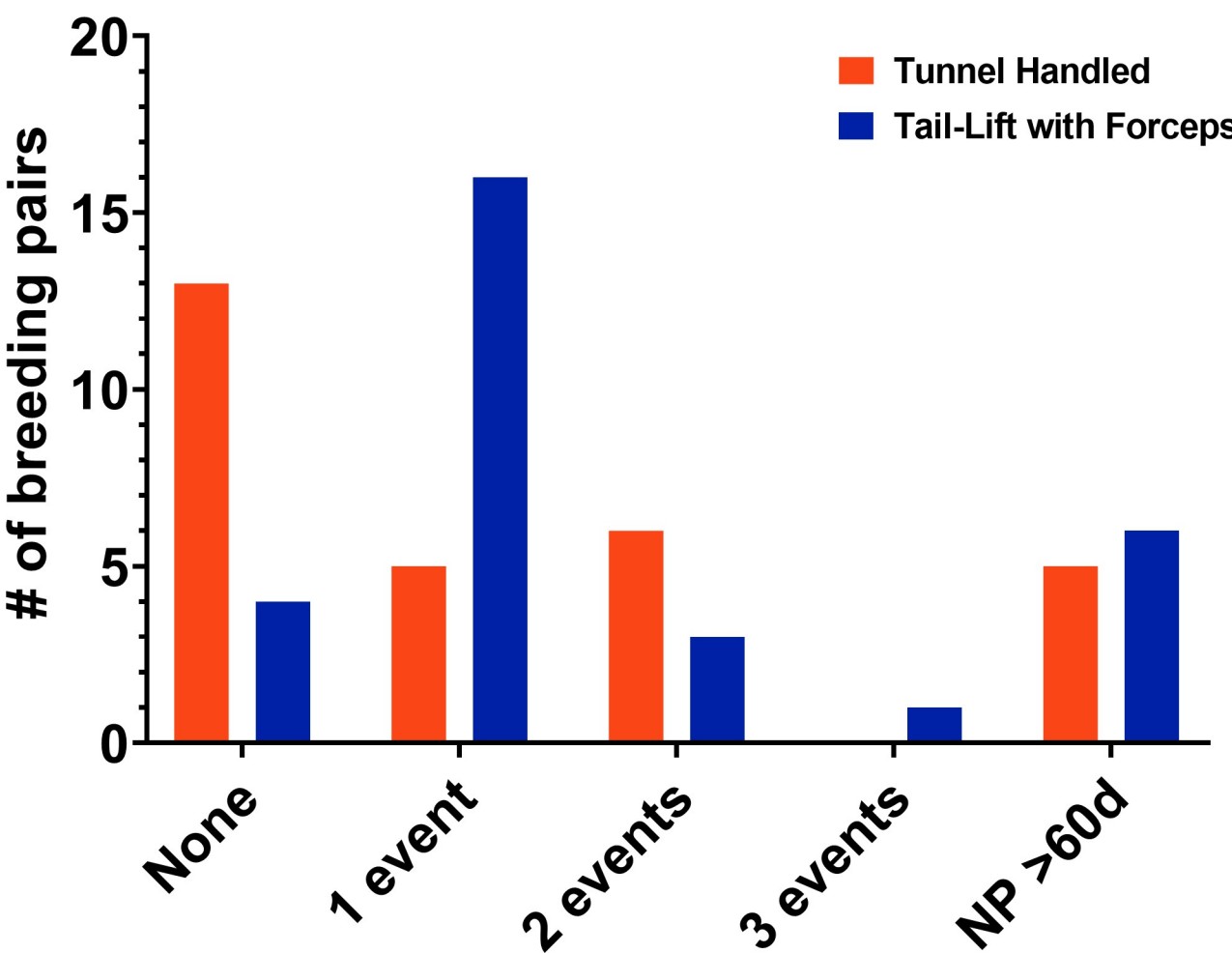

**Fig 2. Frequency of complete litter loss and pair non-productivity.** Complete litter loss events, as defined as disappearance of the entire litter before weaning due to pup death or disappearance, and pair non-productivity defined as no litter produced for >60 days past the previous litter weaned. Orange group = tunnel-handled pairs (n = 29); blue group = tail-lift with forceps handled pairs (n = 30).

exceed 30% in breeding colonies of C57BL/6J mice [28, 29]. The proportion of litters lost is of concern for animal health and welfare reasons [28], and in addition represents a considerable economic sunk cost for large laboratory colonies. In this study, colony litter losses were ~20%, somewhat lower than those previously reported for C57BL/6J mice [29]. We found that litter loss was strongly associated with handling method, with tunnel handled pairs being less likely to lose an entire litter prior to weaning and more likely to successfully wean some pups from each litter produced. Preweaning pup losses may result from a variety of stressors acting on the dam, resulting in poor neonate viability [15, 23, 29]. Attempts to reduce pup mortality have included modifications in availability and type of nesting material [30–32], behavioral enrichment [18], diet [33], husbandry and husbandry-related disturbance [34], and social and housing environment [28]. However, these previous studies did not identify how mice were transferred between cages, so it is not possible to directly compare results with the current study, nor prioritize relative importance of each factor for reducing pup mortality. Maternal experience and parity effects have also been implicated in pup mortality and weaning success, with some studies reporting higher rates of litter loss associated with primiparity [26, 30, 35]. In this study, although all pairs were naïve first-time breeders, loss of whole litters was not

associated with primiparity, as has been observed elsewhere [36]. Approximately 25% of all pups were produced in the first litter, with peak pup and litter loss occurring with the third litter.

Limitations of this study include constraints on space and production, determination of neonate numbers, group differences in home cage configuration, and choice of handling methods. Numbers of breeding pairs available for the study were determined by both facility space and pup demand. Given these constraints, the allowed study size of 60 pairs would have had sufficient power to detect a statistically-significant difference of one extra pup per litter, rather than one extra pup per pair, which was the operational criterion. To detect a difference of one pup per pair would have required a sample size of approximately 210 pairs, which was not feasible given the total size of the operation (S2.1 Text in S2 File). The risk of disturbing the dam shortly after parturition, even with visual checks, meant that numbers of neonates could not always be recorded accurately on the day of birth. It is therefore possible that both litter sizes and pup losses may have been greater than recorded. To reduce the risk cross-cage contamination, tunnels were kept in the home cage of tunnel-handled pairs. In contrast, tail-lift with forceps pairs had no object in their cage so this difference may confound production measurements to some unknown extent. We did not directly compare alternative methods for picking up mice by the tail, such as, by between thumb and forefinger rather than by forceps. There is some evidence that tail-lift by hand may be inherently less stressful and provoke less aggression in male C57BL/6J mice in comparison to forceps handling [37]. In contrast, other studies have shown that mice poorly habituate to tail lift by hand [2, 4]. In this study, we were interested in the comparison of an easily-implemented method of non-aversive handling (tunnels) with the institutional standard of care (tail-lift with forceps), with the goal of assessing the potential for institutional change in practice, rather than evaluating a range of handling methods.

Perceived barriers to full-scale facility implementation of non-aversive handling measures included time required for staff to become familiar with new techniques, investigator resistance to husbandry change, and costs of implementation. Experienced staff were more likely to be resistant to changing accustomed husbandry techniques compared to more recent hires, primarily over procedural time concerns. However, a relatively slow small-scale rollout, and one-on-one staff training, followed by supplementary group training sessions and frequent oversight, enabled correction of procedural problems as they occurred, and demonstrated that cage-change times would be relatively unaffected or even reduced by non-aversive handling. Some investigators were concerned about the potential of a change in handling practice to affect study results. Both group and individual educational sessions were instrumental in persuading researchers that by minimizing animal stress and anxiety with non-aversive handling, results were likely to be more reliable and less variable, thus improving research quality and minimizing the total numbers of animals required. Finally, we demonstrated that the small but consistent increments in pup production associated with tunnel handling could translate to substantial operational revenue and/or cost savings. Startup costs were relatively low; bulk prices of customized and precut tubes purchased from a plastic extrusion company worked out to less than USD $2.00 per tunnel. At an estimated cost of $22 per mouse, and assuming a colony size of 500 breeding pairs, one extra pup born and weaned per pair could result in approximately USD $11,000 in generated income. Alternatively, if at least 30% of tail-lift with forceps pairs are expected to lose at least one entire litter then projected losses could exceed 900 pups, with associated revenue losses of more than USD $20,000. Data from this study and consideration of economic impact were sufficient to persuade management to incorporate non-aversive handling as standard of care throughout the institution. An additional future trial at this facility is now in progress, with the goal of assessing non-aversive handling on productivity of traditionally 'poor' breeding strains, such as BALB/cJ.

Non-aversive handling methods are a simple and inexpensive refinement method with well-documented welfare benefits. This study complements the existing evidence base by expanding application of tunnel-handling to laboratory rodent breeding colonies. Improved welfare was demonstrated through modest increases in pup production and reduction of litter losses, indicating associated reductions in stress. By reducing litter losses in particular, non-aversive handling welfare measures may therefore provide substantial economic benefits to breeding operations with only minor increase in investment, with the potential to substantially reduce total study costs. Finally, this study shows that the subtle and often-overlooked routine husbandry and animal care practices may have profound effects on phenotype and behavior, even for a common and well-documented animal model.

## Methods

### Ethical oversight

The study was approved by the University of Florida Institutional Animal Care and Use Committee (IACUC #201803306). The University of Florida is an AAALAC-accredited institution. Animal care was in strict accordance with the recommendations in the *Guide for the Care and Use of Laboratory Animals* of the National Institutes of Health [38].

### Animal housing and husbandry

The study was conducted at a single academic center on an in-house mouse breeding colony monitored over one breeding cycle of 6 months. The C57BL/6J mice (JAX stock #000664) used in this study were part of an in-house breeding core colony maintained at the University of Florida. The study colony was maintained at room temperature of 23˚C (range ± 2˚C), 30–70% humidity, and a 14:10-h light:dark cycle. Rooms were ventilated with 18 complete positive pressure air changes per hour. Colonies are routinely monitored and maintained negative for 23 known rodent pathogens (viruses, bacteria, fungi, parasites). Operational details are given in S2.2 Text in S2 File.

Mice were housed in pairs in autoclaved JAG75 individually ventilated cages (484 cm$^2$ floor area) with micro-barrier tops (Allentown Caging, Allentown, NJ). Cages were situated on a 140-unit double-sided rack (MicroVent, Allentown Caging, Allentown, NJ). The caging system provided 60 air changes per hour at 22˚C (range ± 1˚C) and 30–40% humidity. Cages were bedded with quarter-inch corn cob bedding (7097, Envigo, Indianapolis, IN), and a compressed autoclaved cotton square for nesting material (Nestlet, Lab Supply, Fort Worth, TX). Food (irradiated rodent diet 2919, Teklad Extruded Diet, Envigo, Indianapolis, IN) and water (reverse-osmosis purified municipal water; Edstrom automatic watering system; Avidity Science, Waterford, WI) were provided *ad libitum*. Cages with weaning-age pups were also provided with a water bottle until their first cage change.

Cage bottoms were changed once every two weeks as part of routine institutional operating procedures. Complete cage changes (including cage tops and water bottles, where used) were performed every 12 weeks during rack change. To minimize disturbing the dam, cages were not opened or changed for 7 days following parturition. If a scheduled cage change fell within this window, it was performed on the 8th day following parturition, or next working day after the 8$^{th}$ day.

### Cage transfer

Animals were transferred to clean cages using one of two methods: tail-lift with forceps or non-aversive handling (tunnels). Control mice were transferred to clean cages by tail-lift with

forceps, the institutional standard technique. Rubber-tipped forceps were used to gently grasp the base of the tail, and the mouse was lifted from the dirty cage and transferred into a clean cage. A small amount of used nesting material was also transferred to maintain a familiar odor environment [39–41]. Between uses, forceps were stored in liquid disinfectant (Peroxigard; Virox Technologies, Oakville, Ontario, Canada). Forceps were alternated between transfers to allow for adequate germicidal contact time.

Animals assigned to experimental cages (tunnels) were transferred using transfer tunnels [4]. The transfer tunnel was an 8.89 x 6.35 x 5.08 cm clear medical-grade polycarbonate square tube (part #J-1002, Petro Extrusion Technologies, Middlesex, NJ), placed in the cage at the time of pairing. During cage transfer, each mouse was gently guided by hand into the tunnel the tunnel was lifted out of the cage and placed inside a clean cage. The tunnel was then gently tipped and the animal was allowed to exit the tunnel. The tunnel stayed in the cage unless visibly soiled. Tunnels were replaced quarterly with new autoclaved tunnels during complete cage changes. A small amount of used nesting material was also transferred to the new cages as was done for the tail-lift with forceps cages.

Cage changes were performed by a team of four husbandry technicians and two breeding colony management technicians. Preliminary training of staff in handling methods was by video instruction (https://www.nc3rs.org.uk/mouse-handling-video-tutorial) followed by in-person training. Staff were periodically checked for correct technique during colony management and were confirmed to be correctly trained at the completion of the project.

## Breeding management

Nulliparous females were monogamously paired with naïve males, 6–8 weeks old. Breeding pairs were selected from currently available animals in the existing colony, and were previously handled with tail-lift with forceps prior to study enrollment. Pairs were maintained together until the end of the breeding lifespan (6 months) to allow for continuous breeding. Pregnancy was checked visually daily via physical appearance without disturbing the cage and at cage change. Parturition was dated from first observation of neonatal pups. Total pup count was obtained within 3 days of birth by visual examination without opening the cage or disturbing the nest. Pups were weaned at approximately 21 days. Weaned pups were distributed for sale to institutional investigators or donated for training of personnel in the internal Animal Care Services unit.

## Experimental design

This was a two-arm randomized controlled trial. Interventions could not be concealed from technical staff, but outcome assessors and the data analyst were masked to allocation. The unit of analysis was the breeding pair. Breeding pairs were randomly allocated 1:1 to transfer method (tail-lift with forceps or tunnel-handled) in blocks of four, for two replicates per block. To minimize potential environmental effects of cage placement on breeding productivity, each block of four cages was also randomized to cage position on the cage rack, using a replicated 2 x 2 Latin square design [42].

The randomization plan was generated in SAS *proc plan* (SAS v.9.4, SAS Inc., Cary NC). Pairing and enrollment were on a rolling basis over three months, to allow flexibility in both meeting anticipated demand and preventing pup over-production.

Breeding pairs were monitored for approximately 180 d (6 months) from date of first pairing. Breeding productivity per pair was assessed by number of pups born per litter, number of pups weaned per litter, inter-litter interval (time between litters, days), and litter loss (all pups died or disappeared before weaning). Breeding efforts were classified as unsuccessful if any of

the following criteria were met: (1) disappearance or loss of an entire litter before weaning, (2) pups were unable to be weaned by d 21 (no incisors, pup too small to reach a wire bar feeder on its own, unthrifty), or (3) dams were found dead or euthanized by $CO_2$ asphyxiation and cervical dislocation for dystocia. Individual pups that were unable to be weaned by d 21 (no incisors, pup too small to reach a wire bar feeder on its own, unthrifty) were euthanized by $CO_2$ asphyxiation and cervical dislocation. Breeding pairs were removed from the study if there were ≥3 instances of complete litter loss, or no litters were produced for more than 60 d from the date of the last litter born (non-productive). Because analysis was intention to treat, data for non-productive pairs or pairs euthanized for cause were included in analysis.

## Statistical analysis

All statistical calculations were performed in SAS (v. 9.4, SAS Inc, Cary NC). Study numbers were constrained by production demand and space constraints to a total sample size of 60 (30 breeding pairs per arm), for a projected total pup production of approximately 1500–2160 pups.

Facility management determined *a priori* that the operationally relevant difference between groups required to demonstrate 'benefit' and justify a transition in handling procedures would be a 5% increase over baseline production. We estimated the operationally significant difference between methods would be approximately one extra pup weaned over the reproductive lifetime of each pair, assuming an expected average litter size of 6 (Mouse Genome Informatics: http://www.informatics.jax.org/), a total of 30 pups over the breeding lifetime of each pair, and baseline weaning success of 75%. A statistically-significant increase in production was estimated to be approximately one extra pup per litter, or 4–5 pups additional per pair. Additional information can be found in S2.3 Text in S2 File.

Descriptive summary statistics of group data were calculated as counts (percentages) and medians (interquartile ranges, IQR). Breeding pair was the unit of analysis. Data were analyzed by intention to treat. The effect of transfer method on breeding metrics was estimated by two-level hierarchical generalized linear mixed models [43–45], with litters nested within breeding pair, transfer method as a fixed effect, and breeding pair as a random effect. The repeated measures structure of litter parity within pair was modelled by an autoregressive AR(1) covariance structure. Models were fitted using residual pseudolikelihood estimation [46] in SAS *proc glimmix*, with best fit assessed by residual plots and diagnostic statistics. Number of pups produced was modelled assuming a Poisson distribution for count outcomes. Number of pups weaned was modelled assuming negative response binomial distribution (SAS *proc glimmix*) to account for overdispersion resulting from zero weaning events. Differences between groups for inter-litter intervals were estimated by mixed model repeated-measures analysis in SAS *proc mixed*, fitted by maximum likelihood estimation. Adverse events per transfer method (litter loss, non-productivity) were assessed by z-tests and likelihood ratio $\chi^2$ test of association (SAS *proc freq*). The hazard ratio for effect of transfer method on recurrent litter losses was estimated in SAS *proc phreg* by the Prentice-Williams-Peterson stratified counting process (CP) model (an extension of the Cox proportional hazards model) for recurrent event data [47–49]. Results are reported as means and 95% confidence intervals.

## Supporting information

**S1 File. ARRIVE 2.0 checklist.**
(PDF)

**S2 File.**
(DOCX)

## Acknowledgments

Special thanks to the UF ACS Breeder Core, colony manager Robert Haynes, Heather Bonanno, Carly Batson, Anita Roberti, and Drs. Brooke Bloomberg & Karl Andrutis for technical and clinical assistance during this project.

## Author Contributions

**Conceptualization:** Margaret A. Hull, Elizabeth A. Nunamaker.

**Data curation:** Margaret A. Hull, Penny S. Reynolds, Elizabeth A. Nunamaker.

**Formal analysis:** Penny S. Reynolds.

**Funding acquisition:** Elizabeth A. Nunamaker.

**Investigation:** Margaret A. Hull, Elizabeth A. Nunamaker.

**Methodology:** Margaret A. Hull, Penny S. Reynolds.

**Project administration:** Margaret A. Hull, Elizabeth A. Nunamaker.

**Resources:** Margaret A. Hull, Elizabeth A. Nunamaker.

**Software:** Penny S. Reynolds.

**Supervision:** Penny S. Reynolds, Elizabeth A. Nunamaker.

**Validation:** Penny S. Reynolds.

**Visualization:** Penny S. Reynolds.

**Writing – original draft:** Margaret A. Hull.

**Writing – review & editing:** Penny S. Reynolds, Elizabeth A. Nunamaker.

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
