## [Decision Letter · Decision Letter 0]

24 Nov 2021

PONE-D-21-29764Effects of non-aversive versus tail-lift handling on breeding productivity in a C57BL/6J mouse colonyPLOS ONE

Dear Dr. Hull,

Thank you for submitting your manuscript to PLOS ONE. After careful consideration, we feel that it has merit but does not fully meet PLOS ONE’s publication criteria as it currently stands. Therefore, we invite you to submit a revised version of the manuscript that addresses the points raised during the review process.

This paper has been reviewed by an independent reviewer as well as by the undersigned editor. In addition to my general expertise in animal welfare that I always apply in my role as editor, I have relied on my specific subject expertise on pup mortality in laboratory animal breeding.

 The paper is generally well written and present relevant results, but requires revision to address the issues detailed in the feedback at the bottom of this message.

We look forward to receiving your revised manuscript.

Kind regards,

I Anna S Olsson, Ph.D.

Academic Editor

PLOS ONE

Journal Requirements:

Additional Editor Comments:

1. Please include line references in the revised version, and refer to these in your response letter.

2. This paper applies different terms for the handling techniques than most papers in the field. Handling mice using a tunnel is usually referred to as tunnel handling, so please use that term here. Standard handling in most studies of handling method is referred to as tail handling / tail picking. However, as far as I know I have never seen tail handling being reported to be done using forceps, so your method seems to be a little different here. Perhaps it can be referred to as forceps tail picking.

3. Page 3 last sentence (the one that continues on page 4): please add a reference for this statement.

4. Page 6, last paragraph: Can you separate litters disappearing from litters having to be euthanized and provide separate figures? Please provide more information on the situations of euthanasia - what was the reason to decide to euthanize a litter? Also, were the three litters from dams found dead or euthanized because of dystocia included in the analysis?

5. Page 7 first paragraph: Please add a definition for recurrent litter loss.

6. Discussion, first paragraph: Does the definition "operationally significant (albeit statistically non-significant) increases" apply generally to the results? You address this further on in the discussion (bottom of page 8), but it becomes a little confusing as you seem to apply more than one definition of operationally significant. See "statistically-significant difference of one extra pup per litter, rather than one extra pup per pair, which was the operational criterion" versus "one extra pup weaned per pair and the magnitude of reduction in litter loss were predetermined to be operationally significant differences". Please see if you can express this in a more straightforward way.

7. Page 8 line 3: There really is very limited support in literature for infanticide being the cause of litter loss. Our most recent paper adds to the evidence that infanticide rarely is the cause when pups die: *Animals*
**2021**, *11*(8), 2327; https://doi.org/10.3390/ani11082327

8. Page 8 middle of the page: The lack of an association between pup mortality and primiparity was also found by us in a retrospective study of breeding data Reproduction in Domestic Animals. doi: 10.1111/j.1439-0531.2012.02147.x. https://onlinelibrary.wiley.com/doi/epdf/10.1111/j.1439-0531.2012.02147.x

9. The standard handling method in your study is tail handling using forceps. I haven't heard of forceps handling in a very long time, and I wonder how generalisable your results are to situations where standard handling is tail handling with the human hand. Please discuss this.

Reviewers' comments:

Reviewer's Responses to Questions

**Comments to the Author**

1. Is the manuscript technically sound, and do the data support the conclusions?

Reviewer #1: Yes

2. Has the statistical analysis been performed appropriately and rigorously? 

Reviewer #1: Yes

3. Have the authors made all data underlying the findings in their manuscript fully available?

Reviewer #1: Yes

4. Is the manuscript presented in an intelligible fashion and written in standard English?

Reviewer #1: Yes

5. Review Comments to the Author

Reviewer #1: Overall comments

I would like to thank the authors for a well written manuscript. Their findings are interesting and novel and have the potential to have significant implications for mouse welfare and economics of mouse breeding colonies. Although their findings remain statistically insignificant, the authors include this caveat accordingly, but present convincing arguments with regards to the economic benefits regardless of the small effect size.

Minor comments/suggestions

The manuscript would benefit from a clearer representation of the data in the forms of its table and figures. Although I appreciate the information portrayed in Table 2, I don’t think this adds to the results section. Furthermore, the findings of one extra pup born and weaned with NAH as outlined in Figure 1 is not immediately clear. Although I appreciate its useful to demonstrate the spread of the data in terms of litter size and occurrence frequency, perhaps the addition of a cumulative number of pups for both would present the findings in a clearer manner. The same applies to the other percentage data presented in the main text, perhaps a summary table of the main findings would aid the reader.

I would appreciate if the authors could include some discussion as to their ‘choice’ or ‘requirement’ for why tail handling is facilitated by forceps as this is not the standard tail handling comparison utilised in most (if not all) the current literature focused on NAH. They could then speculate whether tail handling using forceps is likely to have a greater impact on their measured effects compared to the use of fore finger and thumb.

The authors should also be careful in their phrasing with regards to their data showing direct benefits to animal welfare as they do not have any direct measures of welfare presented in their manuscript (pp 7 discussion line 4). I appreciate that environment (and handling) are important factors of stress which has been shown to influence dam welfare, the fact that there is no data confirming greater stress levels means careful wording is needed.

Methods section – there should be the inclusion of information with regards to the handling of adult mice taken forward for breeding. Were these tail handled up until the point of breeding?

6. PLOS authors have the option to publish the peer review history of their article (what does this mean?). If published, this will include your full peer review and any attached files.

Reviewer #1: **Yes: **Jasmine M Clarkson

---

## [Author Response · Author response to Decision Letter 0]

17 Dec 2021

Additional Editor Comments:

1. Please include line references in the revised version, and refer to these in your response letter.

2. This paper applies different terms for the handling techniques than most papers in the field. Handling mice using a tunnel is usually referred to as tunnel handling, so please use that term here. Standard handling in most studies of handling method is referred to as tail handling / tail picking. However, as far as I know I have never seen tail handling being reported to be done using forceps, so your method seems to be a little different here. Perhaps it can be referred to as forceps tail picking.

 Author response: Acronyms for treatment groups were removed in full throughout the manuscript. Non-aversive handling was changed to “tunnel handling” or “tunnel-handled” where appropriate. Figures 1 & 2 were also adjusted to have consistent labelling. Tail-lift with forceps is a very common (that is nearly ubiquitous) method of cage change handling in the United States and much of North America. This is discussed in more depth in (Doerning 2019), and is now referenced in the manuscript. 

3. Page 3 last sentence (the one that continues on page 4): please add a reference for this statement.

 Author response: Citation added, (Orso, 2017), page 4 line 92. 

4. Page 6, last paragraph: Can you separate litters disappearing from litters having to be euthanized and provide separate figures? Please provide more information on the situations of euthanasia - what was the reason to decide to euthanize a litter? Also, were the three litters from dams found dead or euthanized because of dystocia included in the analysis?

 Author response: Table 1 (page 5 line 124-125) was expanded to include pups euthanized and litters lost. No entire litter was euthanized, only individual pups meeting stated endpoint criteria on page 14 lines 423-432 were euthanized. Added additional clarifying information for exact pup endpoint criteria (page 14 lines 427-429). Litters from dams that were found dead or euthanized for health concerns were included for analysis, as the analysis was conducted as ‘intention to treat’. This was added to page 14 lines 431-432 and page 15 lines 449-450. 

5. Page 7 first paragraph: Please add a definition for recurrent litter loss.

 Author response: Added, page 6 line 168 and page 15 line 449-450. 

6. Discussion, first paragraph: Does the definition "operationally significant (albeit statistically non-significant) increases" apply generally to the results? You address this further on in the discussion (bottom of page 8), but it becomes a little confusing as you seem to apply more than one definition of operationally significant. See "statistically-significant difference of one extra pup per litter, rather than one extra pup per pair, which was the operational criterion" versus "one extra pup weaned per pair and the magnitude of reduction in litter loss were predetermined to be operationally significant differences". Please see if you can express this in a more straightforward way.

 Author response: Reworded page 2 lines 31-33, page 6 lines 133-138.

7. Page 8 line 3: There really is very limited support in literature for infanticide being the cause of litter loss. Our most recent paper adds to the evidence that infanticide rarely is the cause when pups die: Animals 2021, 11(8), 2327; https://doi.org/10.3390/ani11082327

 Author response: We agree and have removed this phrase. 

8. Page 8 middle of the page: The lack of an association between pup mortality and primiparity was also found by us in a retrospective study of breeding data Reproduction in Domestic Animals. doi: 10.1111/j.1439-0531.2012.02147.x. https://onlinelibrary.wiley.com/doi/epdf/10.1111/j.1439-0531.2012.02147.x

 Author response: This citation was added. It was in a previous draft but was inadvertently omitted during revision. Thank you for the reminder. 

9. The standard handling method in your study is tail handling using forceps. I haven't heard of forceps handling in a very long time, and I wonder how generalisable your results are to situations where standard handling is tail handling with the human hand. Please discuss this.

Author response: As stated previously cage change utilizing forceps is very common in the United States, and is discussed more in depth in (Doerning, 2019). We expanded upon this topic on page 8 lines 273-280. This paper intended to compare our previous institutional standard method to tunnels, with the goal of institutional change of practice. 

Reviewers' comments:

Reviewer's Responses to Questions

Comments to the Author

1. Is the manuscript technically sound, and do the data support the conclusions?

Reviewer #1: Yes

2. Has the statistical analysis been performed appropriately and rigorously? 

Reviewer #1: Yes

3. Have the authors made all data underlying the findings in their manuscript fully available?

Reviewer #1: Yes

4. Is the manuscript presented in an intelligible fashion and written in standard English?

Reviewer #1: Yes

5. Review Comments to the Author

Reviewer #1: Overall comments

I would like to thank the authors for a well written manuscript. Their findings are interesting and novel and have the potential to have significant implications for mouse welfare and economics of mouse breeding colonies. Although their findings remain statistically insignificant, the authors include this caveat accordingly, but present convincing arguments with regards to the economic benefits regardless of the small effect size.

Minor comments/suggestions

The manuscript would benefit from a clearer representation of the data in the forms of its table and figures. Although I appreciate the information portrayed in Table 2, I don’t think this adds to the results section. 

 Author response: Table 2 has been moved to supplemental materials, S2.5 Table 1. 

Furthermore, the findings of one extra pup born and weaned with NAH as outlined in Figure 1 is not immediately clear. Although I appreciate its useful to demonstrate the spread of the data in terms of litter size and occurrence frequency, perhaps the addition of a cumulative number of pups for both would present the findings in a clearer manner. 

 Author response: We agree, and have revised figure 1 to show cumulative totals for pups born and weaned per pair. 

The same applies to the other percentage data presented in the main text, perhaps a summary table of the main findings would aid the reader.

Author response: We agree, and have revised table 1 to address reviewer comments, and moved Table 2 and count data to supplemental results, S2.5 Table 1 & 2. 

I would appreciate if the authors could include some discussion as to their ‘choice’ or ‘requirement’ for why tail handling is facilitated by forceps as this is not the standard tail handling comparison utilized in most (if not all) the current literature focused on NAH. They could then speculate whether tail handling using forceps is likely to have a greater impact on their measured effects compared to the use of fore finger and thumb.

Author response: As stated for the previous reviewer, tail-lift with forceps is an extremely common method in the US, and is often the institutional standard due to concern over biosecurity, prevention of cross-contamination, and operator injury (albeit there is evidence showing that this method increases the incidence of operator repetitive injury!). Although we have little hard information as to the actual prevalence of tail handling with forceps in US academic institutions, we have plenty of anecdotal evidence to suggest it is unfortunately a widespread practice in the US. We have provided a reference in support (Doerning, 2019, page 8 lines 273 – 280). 

The authors should also be careful in their phrasing with regards to their data showing direct benefits to animal welfare as they do not have any direct measures of welfare presented in their manuscript (pp 7 discussion line 4). I appreciate that environment (and handling) are important factors of stress which has been shown to influence dam welfare, the fact that there is no data confirming greater stress levels means careful wording is needed.

Author response: Changed “welfare benefits” to “productivity” and added a sentence (page 9, line 232-233) to suggest that welfare benefits associated with the reduction in possibly preventable pup deaths. 

Methods section – there should be the inclusion of information with regards to the handling of adult mice taken forward for breeding. Were these tail handled up until the point of breeding?

Author response: Yes, as it was the institutional method of mouse handling during routine husbandry care at the time, so this is an unavoidable confound. Added a sentence explaining this, (page 13, line 398-399). 

---

## [Decision Letter · Decision Letter 1]

11 Jan 2022

PONE-D-21-29764R1Effects of non-aversive versus tail-lift handling on breeding productivity in a C57BL/6J mouse colonyPLOS ONE

Dear Dr. Hull,

Thank you for submitting your manuscript to PLOS ONE. After careful consideration, we feel that it has merit but does not fully meet PLOS ONE’s publication criteria as it currently stands. Therefore, we invite you to submit a revised version of the manuscript that addresses the points raised during the review process.

The manuscript has been thoroughly revised and all comments in the previous round of reviews have been addressed, and there is only one minor issue that need clarifying:On lines 163-164 and on lines 175-176, you write about an increase in the proportion of all litters successfully weaned and a decrease in likelihood that an entire litter was lost prior to weaning. Can you please clarify how these results are independent? I may be misunderstanding this, but it seems to me that either a litter is successfully weaned or it is lost prior to weaning, and therefore these two results are dependent on each other, which would make it redundant to report both.

We look forward to receiving your revised manuscript.

Kind regards,

I Anna S Olsson, Ph.D.

Academic Editor

PLOS ONE

Journal Requirements:

Reviewers' comments:

Reviewer's Responses to Questions

**Comments to the Author**

1. If the authors have adequately addressed your comments raised in a previous round of review and you feel that this manuscript is now acceptable for publication, you may indicate that here to bypass the “Comments to the Author” section, enter your conflict of interest statement in the “Confidential to Editor” section, and submit your "Accept" recommendation.

Reviewer #1: All comments have been addressed

2. Is the manuscript technically sound, and do the data support the conclusions?

Reviewer #1: Yes

3. Has the statistical analysis been performed appropriately and rigorously? 

Reviewer #1: Yes

4. Have the authors made all data underlying the findings in their manuscript fully available?

Reviewer #1: Yes

5. Is the manuscript presented in an intelligible fashion and written in standard English?

Reviewer #1: Yes

6. Review Comments to the Author

Reviewer #1: I would like to thank the authors for their revisions to the manuscript. The manuscript is now much clearer with regards to their findings and revision to the figures aids clarity of their findings.

7. PLOS authors have the option to publish the peer review history of their article (what does this mean?). If published, this will include your full peer review and any attached files.

Reviewer #1: **Yes: **Jasmine M Clarkson

---

## [Author Response · Author response to Decision Letter 1]

11 Jan 2022

Tuesday, January 11, 2022

The manuscript has been thoroughly revised and all comments in the previous round of reviews have been addressed, and there is only one minor issue that need clarifying:

On lines 163-164 and on lines 175-176, you write about an increase in the proportion of all litters successfully weaned and a decrease in likelihood that an entire litter was lost prior to weaning. Can you please clarify how these results are independent? I may be misunderstanding this, but it seems to me that either a litter is successfully weaned or it is lost prior to weaning, and therefore these two results are dependent on each other, which would make it redundant to report both.

Author response: We agree. We have altered the language on these sentences to be more clear when discussing all litters produced by a single dam and all litters for an entire treatment group.

---

## [Editor Report · Decision Letter 2]

14 Jan 2022

Effects of non-aversive versus tail-lift handling on breeding productivity in a C57BL/6J mouse colony

PONE-D-21-29764R2

Dear Dr. Hull,

We’re pleased to inform you that your manuscript has been judged scientifically suitable for publication and will be formally accepted for publication once it meets all outstanding technical requirements.

Kind regards,

I Anna S Olsson, Ph.D.

Academic Editor

PLOS ONE
---

## [Editor Report · Acceptance letter]

19 Jan 2022

PONE-D-21-29764R2 

Effects of non-aversive versus tail-lift handling on breeding productivity in a C57BL/6J mouse colony 

Dear Dr. Hull:

I'm pleased to inform you that your manuscript has been deemed suitable for publication in PLOS ONE. Congratulations! Your manuscript is now with our production department. 

Kind regards, 

on behalf of

Dr. I Anna S Olsson 

Academic Editor

PLOS ONE